# Task-specific invariant representation in auditory cortex

Charles R Heller[1], Gregory R Hamersky[1], Stephen V David[2]*

[1]Neuroscience Graduate Program, Oregon Health and Science University, Portland, United States; [2]Otolaryngology, Oregon Health & Science University, Portland, United States

**Abstract** Categorical sensory representations are critical for many behaviors, including speech perception. In the auditory system, categorical information is thought to arise hierarchically, becoming increasingly prominent in higher-order cortical regions. The neural mechanisms that support this robust and flexible computation remain poorly understood. Here, we studied sound representations in the ferret primary and non-primary auditory cortex while animals engaged in a challenging sound discrimination task. Population-level decoding of simultaneously recorded single neurons revealed that task engagement caused categorical sound representations to emerge in non-primary auditory cortex. In primary auditory cortex, task engagement caused a general enhancement of sound decoding that was not specific to task-relevant categories. These findings are consistent with mixed selectivity models of neural disentanglement, in which early sensory regions build an overcomplete representation of the world and allow neurons in downstream brain regions to flexibly and selectively read out behaviorally relevant, categorical information.

*For correspondence: davids@ohsu.edu

Competing interest: The authors declare that no competing interests exist.

## eLife assessment

This **important** study provides insights into how the brain constructs categorical neural representations during a difficult auditory target detection task. Through recordings of simultaneous single-unit activity in primary and secondary auditory areas, **compelling** evidence is provided that categorical neural representations emerge in a secondary auditory area, i.e., PEG. The study is of interest to neuroscientists and can also potentially shed light on human psychological studies.

## Introduction

Perceptual decision-making requires behavioral responses based on specific sensory patterns that ignore distracting and irrelevant information. In the auditory system, categorical sensory representation is essential to many natural behaviors (*Bizley and Cohen, 2013*). For example, during language processing, vowels are perceived categorically, even though the formant frequencies that define them vary continuously across utterances (*Hillenbrand et al., 1995*). Categorical perception is not limited to language, as subjects can learn to classify arbitrary, novel sounds according to one spectro-temporal feature while ignoring others (*Stilp and Kluender, 2010*).

Neurophysiological studies in auditory cortex have shown that engaging in auditory behavior can enhance sensory discriminability at the level of single neurons (*Buran et al., 2014*; *Niwa et al., 2012a*) and neural populations (*Bagur et al., 2018*; *Kuchibhotla et al., 2017*). Most of this work has demonstrated a generalized, overall improvement in sensory coding without contrasting neural representations of task-relevant versus -irrelevant features. In frontal cortex, neurons often only encode sound categories (*Fritz et al., 2010*; *Tsunada et al., 2011*), suggesting that sound information is transformed into an invariant, categorical representation before exiting auditory cortex. Such

representations require disentangling sensory features that are relevant for defining the object category from other features that are irrelevant to the category (***DiCarlo and Cox, 2007***). Theory predicts that neural systems can produce these invariant representations through hierarchical computation. In early processing regions, mixed selectivity of single neurons produces high-dimensional, overcomplete representations of sensory inputs and behavioral variables. From this population activity, it is straightforward for neurons in downstream areas to decode information about a specific feature that is important to the current behavior and whose representation is invariant to irrelevant sensory information (***Kell et al., 2018***; ***Rigotti et al., 2013***).

We hypothesized that invariant auditory representations supporting perceptual discrimination arise through a behavior-dependent hierarchical process, consistent with mixed selectivity models. According to this model, engaging in an auditory behavior leads to a non-specific enhancement of auditory representations at early stages, followed by a selective enhancement of task-relevant features at later stages. Previous work has shown that the effects of task engagement are larger in non-primary auditory fields (***Atiani et al., 2014***; ***Kline et al., 2023***; ***Niwa et al., 2013***), as are the effects of selective attention, which may be related to invariant sound feature coding (***O'Sullivan et al., 2019***). Some studies have also reported that choice-related activity emerges in non-primary auditory cortex during a challenging perceptual discrimination behavior (***Tsunada et al., 2016***), although factors affecting choice coding may be task-dependent (***Bizley et al., 2013***). Together, these findings are consistent with the idea that behaviorally relevant neural representations are computed hierarchically in auditory cortex (***Lestang et al., 2023***).

To investigate the emergence of invariant sound coding, we recorded neural population activity from primary and non-primary fields of the ferret auditory cortex while animals alternated between active tone-in-noise detection and passive listening to task stimuli. We designed the task so that behavioral sessions contained multiple different target and distractor sounds and used decoding analysis to measure how neural populations discriminate between both task-relevant and -irrelevant

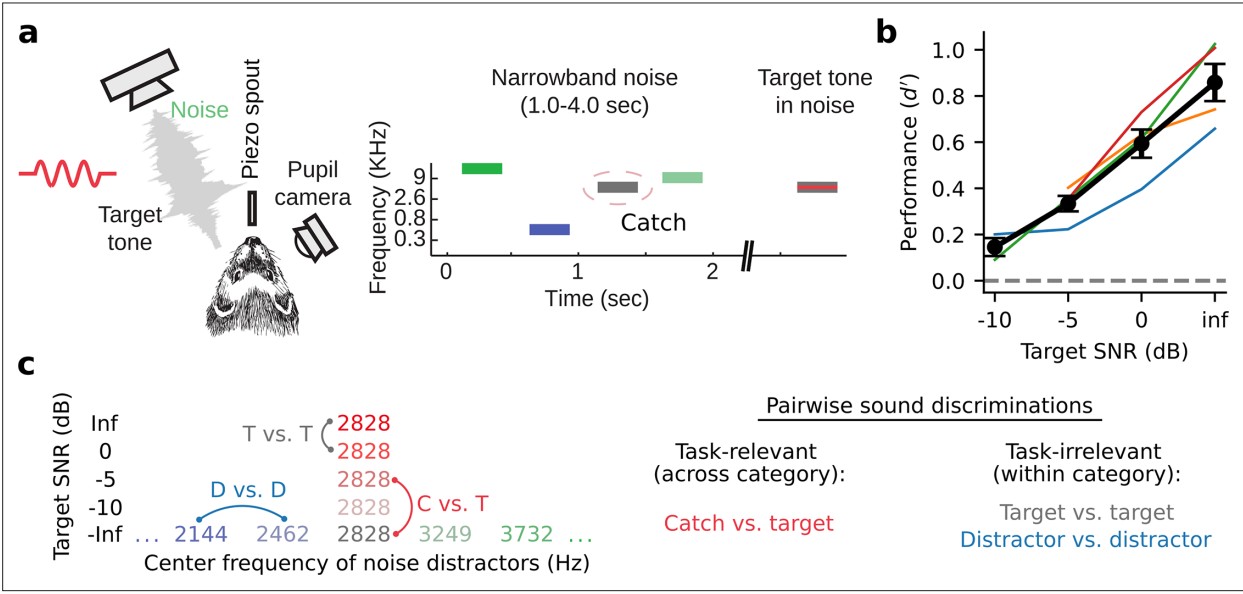

**Figure 1.** Tone-in-noise detection behavior. (**a**) Schematic of go/no-go tone-in-noise detection task. Licking responses to target tones were rewarded, while responses to narrowband noise distractors were penalized with a timeout. Target tone frequency was fixed during a single behavior session and masked by narrowband (0.3 octave) noise centered at the same frequency with variable signal-to-noise ratio (SNR). Variable SNR was achieved by varying the overall SPL of the target relative to the fixed (60 dB SPL) distractor noise, e.g., –5 dB SNR corresponds to a 55 dB SPL target with 60 dB SPL masking noise. Infinite (inf) dB SNR corresponds to a target tone presented in isolation (60 dB SPL). The 'Catch' distractor was identical to the masking noise but with no tone. (**b**) Behavioral performance of individual animals as a function of SNR (*d-prime=Z*[target response rate] - *Z*[catch response rate], n=4 animals). Black lines and error bars indicate the mean and standard error of the mean across animals. (**c**) Left: Stimulus set for an example experiment where the target tone frequency was 2828 Hz. Right: both task-relevant (catch vs. target) and task-irrelevant (target vs. target, distractor vs. distractor) sound discriminations were studied.

The online version of this article includes the following figure supplement(s) for figure 1:

**Figure supplement 1.** Behavioral performance of individual animals.

sound features. For this analysis, we developed decoding-based dimensionality reduction (dDR), which projects neural activity into a low-dimensional subspace spanning both changes in mean firing rate between categories and covariability across trials (*Heller and David, 2022*) dDR prevents bias that can affect population decoding in behavioral studies with relatively limited numbers of trials (*Kanitscheider et al., 2015*; *Moreno-Bote et al., 2014*). Effects of task engagement were highly variable across individual neurons, but the population-level analysis revealed that sound coding in primary auditory cortex was broadly and non-specifically improved by task engagement. In contrast, an enhanced, selective representation of task-relevant features emerged in non-primary auditory cortex. The degree of task-relevant enhancement was correlated with behavioral performance, consistent with the hypothesis that categorical representations in non-primary auditory cortex inform behavioral choices.

## Results

### Psychometric tone-in-noise detection behavior

To study how neural representations of sound category emerge in auditory cortex, we trained four male ferrets to perform a go/no-go tone-in-noise detection task. Animals reported the occurrence of a target tone in a sequence of narrowband noise distractors by licking a piezo spout (*Figure 1A*, Methods: Behavioral paradigm, distractor stimulus sound level: 60 dB SPL). Targets were presented with variable signal-to-noise ratio (SNR), masked by noise centered at the same frequency. We describe SNR as the overall SPL of the target relative to the distractor noise level. Thus, an SNR of –5 dB corresponds to a target level of 55 dB SPL while an Inf dB SNR corresponds to a target tone presented without any masking noise. A subset of behavioral trials in each experiment included an explicit catch stimulus whose center frequency was matched to that of the target tone. This task design permitted us to probe neural coding of both task-relevant sound features (presence or absence of a target tone) and -irrelevant features (level of noise masking the target tone, *Figure 1C*).

Behavioral performance was measured using d-prime (*Green and Swets, 1966*; *Saderi et al., 2021*), calculated as the z-scored hit rate for a given target minus the z-scored catch response rate. Across behavioral sessions and animals, performance showed a clear psychometric dependence on target SNR (*Figure 1B*, *Figure 1—figure supplement 1*). All animals could easily discriminate between the pure tone (Inf dB) target and catch stimulus, while performance for lower SNR target stimuli approached the chance level.

### Diverse effects of task engagement on single neurons in primary and non-primary auditory cortex

We used linear 64-channel silicon probes (*Shobe et al., 2015*) to record single-unit activity from primary (A1) and non-primary (dPEG) auditory cortex while animals performed the tone-in-noise detection task. Recordings were targeted to each respective region based on functional mapping of neural responses (Methods, *Figure 2—figure supplement 1*). Behavior alternated between blocks of active task engagement and passive listening to task stimuli. During passive listening, licking responses were not rewarded and animals quickly disengaged from the task (*David et al., 2012*).

Sound-evoked spiking activity was compared between active and passive states to study the impact of task engagement on sound representation. In both A1 and dPEG, responses to target and catch stimuli were significantly discriminable for a subset of single neurons (about 25% in both areas, *Figure 2A–C*, *Figure 2—figure supplements 2–4*, bootstrap test). This supports the idea that stimulus identity can be decoded in both brain regions, regardless of task performance. However, the fact that the responses of most neurons in both brain areas could not significantly discriminate target vs. catch stimuli also highlights the diversity of sound encoding observed at the level of single neurons. The accuracy of catch vs. target discrimination for each neuron was quantified using neural d-prime, the z-scored difference in target minus catch spiking response for each neuron (Methods: Single neuron PSTHs and d-prime *Niwa et al., 2012a*). Task engagement was associated with significant changes in catch vs. target d-prime for roughly 10% of neurons in both A1 (40/481 neurons, bootstrap test) and dPEG (33/377 neurons, bootstrap test). This included neurons that both increased their discriminability and decreased their discriminability (*Figure 2D–E*). Thus, the effects of task engagement at

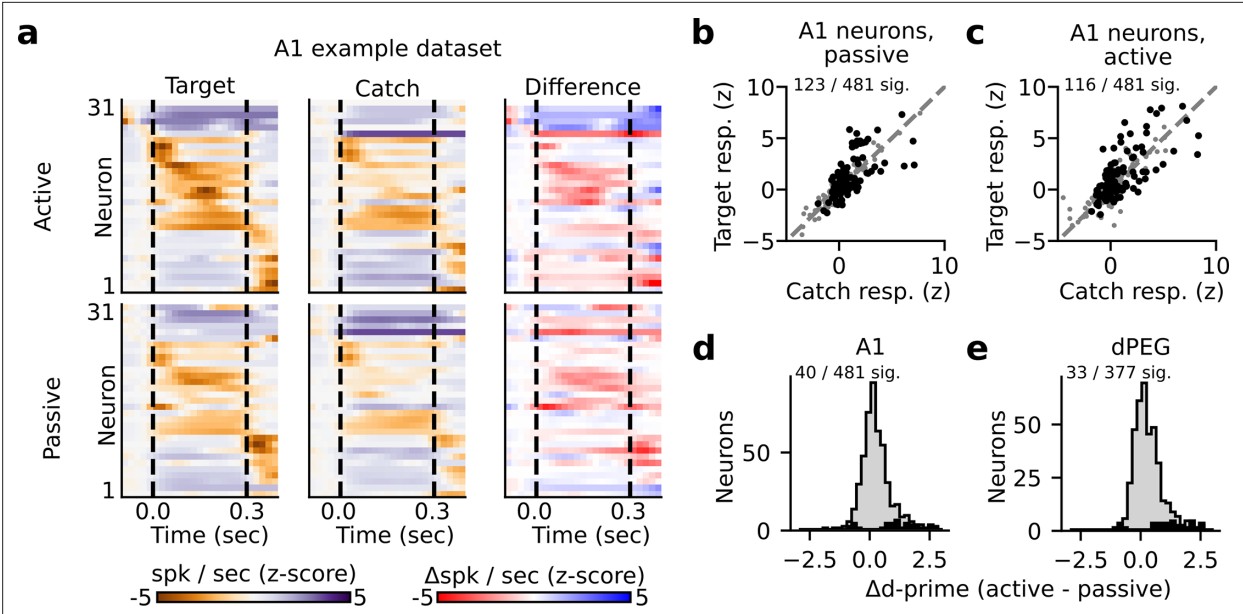

**Figure 2.** State-dependent modulation of single neuron target vs. catch discrimination. (**a**) Example peristimulus time histogram (PSTH) responses from a single recording site in A1. Heatmap color in each row indicates the PSTH amplitude of one neuron. Dashed lines indicate sound onset/offset. Spikes were binned (20 ms), z-scored, and smoothed (σ=30 ms Gaussian kernel). Example target responses are to the pure tone (Inf dB) target. Difference is computed as the z-scored response to the target minus the z-scored catch response (resulting in a difference shown in units of z-score). (**b–c**) Mean z-scored response evoked by-catch vs. Inf dB stimulus for each A1 neuron (n=481 neurons) across passive (**b**) and active (**c**) trials. Responses were defined as the total number of spikes recorded during the 300ms of sound presentation (area between dashed lines in panel A). Neurons with a significantly different response to the catch vs. target stimulus are indicated in black and quantified on the respective figure panel. (**d**) Histogram plots the state-dependent change in target vs. catch stimulus discriminability for each A1 neuron. Neural d-prime is defined |Z[target] - Z[catch]|, and Δd-prime is the difference of active minus passive d-prime. The distribution of neurons with a significant change in d-prime between passive and active conditions is overlaid in black. (**e**) Histogram of Δd-prime for dPEG neurons, plotted as in D.

The online version of this article includes the following figure supplement(s) for figure 2:

**Figure supplement 1.** Penetration map for one example animal.

**Figure supplement 2.** State-dependent modulation of singe neuron target vs. catch discrimination in dPEG.

**Figure supplement 3.** Single neuron target vs. catch raster plots for all A1 recording sites.

**Figure supplement 4.** Single neuron target vs. catch raster plots for all dPEG recording sites.

the level of single neurons were relatively mild and inconsistent across the population; many neurons showed no significant change, and of those that did, the effects were bidirectional (*Figure 2D–E*).

## Population coding of task-relevant features is selectively enhanced in non-primary auditory cortex

Given the diversity of task-related changes in neural activity, we asked if a clearer pattern of task-dependent changes could be observed at the population level. We performed optimal linear decoding of task stimuli from the single-trial activity of simultaneously recorded neurons at each recording site. We quantified decoding performance with neural population d-prime (Methods: Neural population d-prime *Abbott and Dayan, 1999*; *Niwa et al., 2012a*). To prevent overfitting and allow visualization of population responses, we first projected single trial activity into a low-dimensional subspace optimized for linear decoding of task stimuli (*Figure 3*; *Heller and David, 2022*). In both A1 and dPEG, population d-prime for catch versus target stimuli consistently increased during task engagement. In A1, the increase in d-prime was consistent across all task categories; there was no difference between target vs. target and target vs. catch discrimination accuracy (*Figure 3B–C*). However, in dPEG the improvement of task-relevant catch vs. target discrimination was significantly larger than in any other category (*Figure 3E–F*). Unlike A1, discrimination of task-relevant sound categories was selectively enhanced in non-primary auditory cortex.

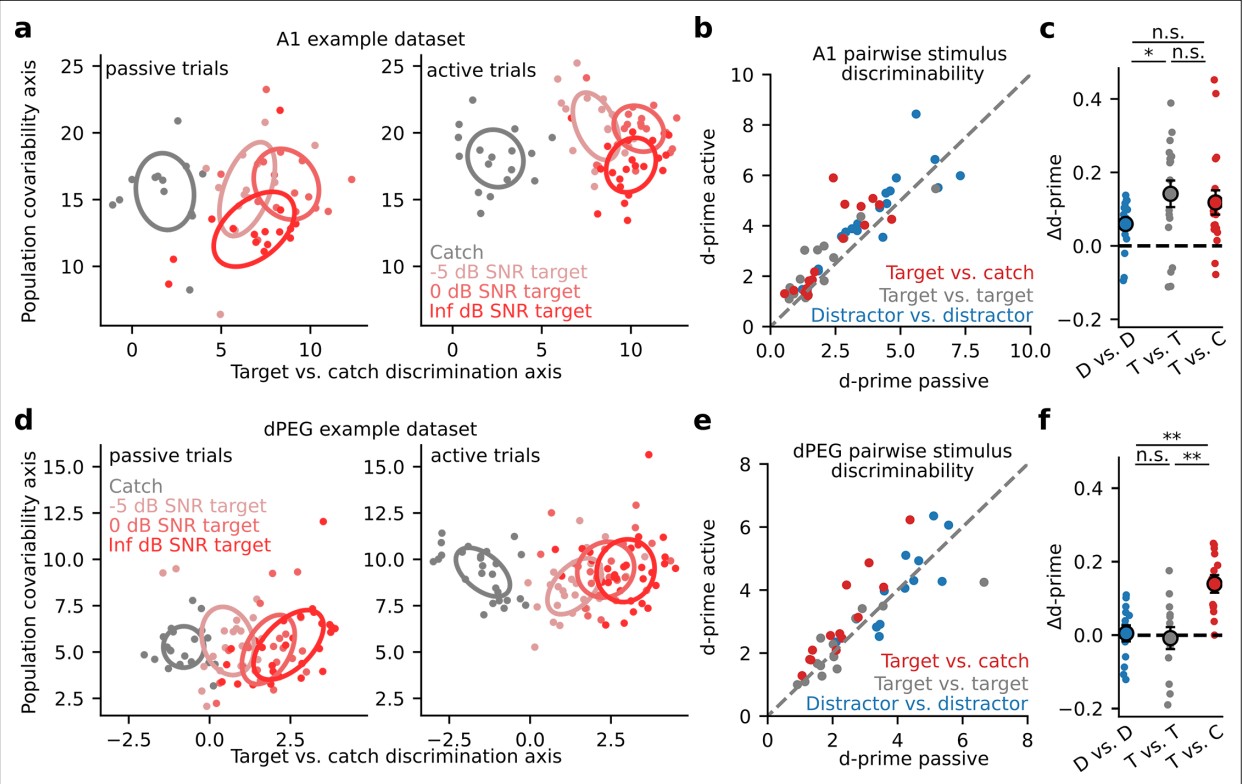

**Figure 3.** Selective enhancement of task-relevant category representation in the secondary auditory cortex. (**a**) Left: Representative A1 population activity during passive listening projected into a two-dimensional space optimized for discriminating target versus catch responses. Each dot indicates the population response on a single trial, color indicates different noise (catch) or tone-in-noise (target) stimuli, and ellipses describe the standard deviation of responses across trials. The degree of ellipse overlap provides a visualization of the neural discriminability (d-prime) between the corresponding stimuli. Right: A1 population activity during active behavior. (**b**) Mean population d-prime between sounds from each category (target vs. catch, target vs. target, and distractor vs. distractor, *Figure 1C*) for each A1 recording site (n=18 sessions, n=3 animals). (**c**) Δd-prime is the difference between active and passive d-prime, normalized by their sum (D vs. D / T vs. T p=0.048, Wilcoxon signed-rank test). Each small dot represents the mean for a single A1 recording site, as in panel b (n=18 sessions, n=3 animals). Large dots and error bars represent the mean and standard error across sessions. (**d**) Single-trial population responses for a single site in non-primary auditory cortex (dPEG), plotted as in A. (**e**) Passive vs. Active category discriminability for dPEG recording sites, plotted as in B (n=12 sessions, n=4 animals). (**f**) Data shown as in panel c but for changes in discriminability per category in dPEG (n=12 sessions, n=4 animals). Δd-prime for target vs. catch pairs (T vs. C) was significantly greater than for the other categories (D vs. D: p=0.003; T vs. T: p=0.005, Wilcoxon signed-rank test).

The online version of this article includes the following figure supplement(s) for figure 3:

**Figure supplement 1.** Pupil dynamics reflect both generalized arousal as well as trial outcome.

**Figure supplement 2.** Selective enhancement of task-relevant category representation in the secondary auditory cortex is not affected by global arousal.

Prior work has demonstrated that generalized, pupil-indexed arousal can impact the responses of neurons in auditory cortex, independent of engagement in a specific task (*McGinley et al., 2015*; *Schwartz et al., 2020*). Importantly, task engagement is often correlated with increased arousal (*de Gee et al., 2022*; *Saderi et al., 2021*; *Figure 3—figure supplement 1*). To ensure that our results were not influenced by non-specific effects of arousal, decoding analysis was performed after first removing all spiking variability that could be explained using pupil-indexed arousal (Methods). Performing the same decoding analysis without first controlling for pupil size did not affect the selective enhancement that we observed in dPEG (*Figure 3—figure supplement 2*). However, Δ d-prime, in both A1 and dPEG, was higher overall. The absence of a pupil effect on selectivity suggests that pupil-indexed arousal primarily operates on an orthogonal subspace to the global task engagement axis and tends to non-specifically improve coding accuracy.

In addition to reflecting overall arousal level, pupil size has also been reported to reflect more nuanced cognitive variables such as, for example, listening effort (*Zekveld et al., 2014*). Furthermore,

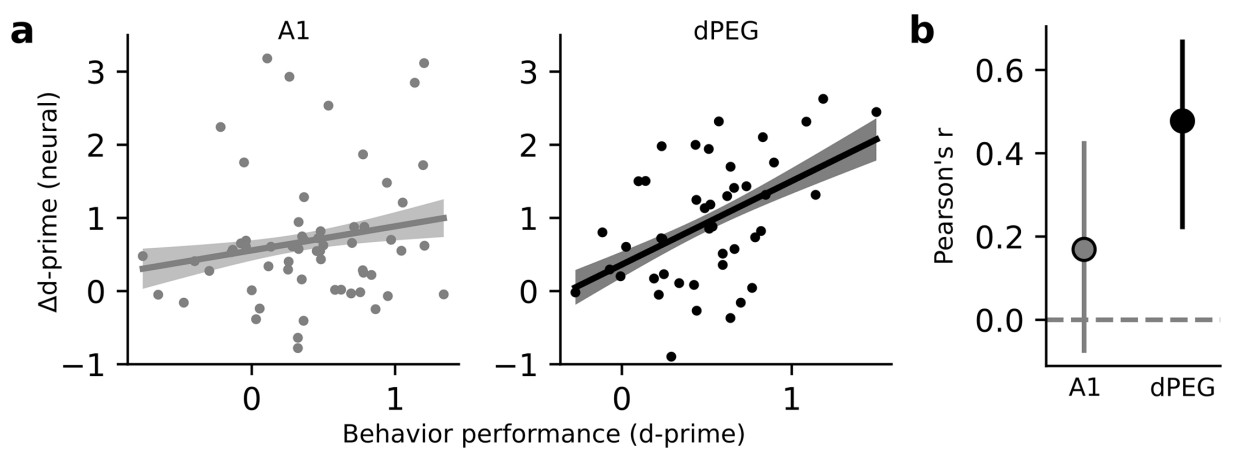

**Figure 4.** Changes in neural decoding are correlated with behavior performance in dPEG, but not A1. (**a**) Scatter plot compares neural Δd-prime (active minus passive) for all tone-in-noise target vs. catch noise combinations against the corresponding behavioral d-prime for that target vs. catch discrimination. Line shows the best linear fit, and shading represents a bootstrapped 95% confidence interval for slope. Left, data from A1 (n=60 unique target vs. catch combinations, n=3 animals, 18 recording sessions). Right, data from dPEG (n=44 unique target vs. catch combinations, n=4 animals, 12 recording sessions). (**b**) Pearson correlation between neural d-prime and behavioral d-prime in each brain region. Error bars indicate bootstrapped 95% confidence intervals (A1: p=0.082; dPEG: p=0.002, bootstrap test).

The online version of this article includes the following figure supplement(s) for figure 4:

**Figure supplement 1.** Choice decoding in auditory cortex primarily reflects impulsivity.

rodent data suggests that optimal sensory detection is associated with intermediate pupil size (*McGinley et al., 2015*), consistent with the hypothesis of an inverted-U relationship between arousal and behavioral performance (*Zekveld et al., 2014*). To determine if this pattern was true for the animals in our task, we measured the dynamics of pupil size in the context of behavioral performance. Across animals, task stimuli evoked robust pupil dilation that varied with the trial outcome (*Figure 3—figure supplement 1b–c*). Notably, pre-trial pupil size was significantly different between correct (hit and correct reject), hit, and miss trials (*Figure 3—figure supplement 1b–c*), recapitulating the finding of an inverted-U relationship to performance in rodents (*McGinley et al., 2015*). Since we focused only on correct trials in our decoding analysis, these outcome-dependent differences in pupil size are unlikely to contribute to the emergent decoding selectivity in dPEG.

### Behavioral performance is correlated with neural coding changes in non-primary auditory cortex only

If task-related changes in neural coding are linked to processes that guide behavior, then the changes in neural activity should be predictive of behavioral performance (*Tsunada et al., 2016*). While selective enhancement of task-relevant discriminability was observed only in dPEG, both areas showed an overall increase in sensory discriminability. We asked if either of these changes in neural decoding performance were predictive of behavioral performance. For each tone-in-noise target stimulus, we compared the task-related change in neural d-prime to behavioral d-prime in the same experiment. We found a significant correlation for populations in dPEG, but not in A1 (*Figure 4*). Thus, the task-specific changes in dPEG are coupled with the behavioral output reflecting those sound features.

The previous analysis suggests that the task-dependent increase in stimulus information present in dPEG population activity is predictive of overall task performance. Next, we asked whether the population activity in either brain region was directly predictive of behavioral choice on single hit vs. miss trials. To do this, we conducted a choice probability analysis (Methods). We found that in both brain regions choice could be decoded well above chance level (*Figure 4—figure supplement 1*). Choice information was present throughout the entire trial and did not increase during the target stimulus presentation. This suggests that the difference in population activity primarily reflects a cognitive state associated with the probability of licking on a given trial, or 'impulsivity' rather than 'choice.' This interpretation is consistent with our finding that baseline pupil size on each trial is predictive of trial outcome (*Figure 3—figure supplement 1b*).

# Changes in evoked response gain, not shared population covariability, support the emergence of categorical representations in non-primary auditory cortex

The difference in task-dependent coding between A1 and dPEG could be explained by differential changes in the evoked responses of single neurons, patterns of covariability between neurons, or both (*Cohen and Maunsell, 2009*; *Cowley et al., 2020*). To measure task-dependent changes in covariability, we used factor analysis to model low-dimensional correlated activity in the neural population (Methods: Factor analysis). We found that covariability patterns changed significantly between the passive and active states (*Figure 5—figure supplement 1*). In both brain regions, task engagement caused a rotation of the principal covariability axis, consistent with an overall decorrelation of population activity (*Umakantha et al., 2021*). In theory, a rotation could either help, or hurt, decoding accuracy, depending on its alignment with the sound discrimination axis (*Figure 5A*). Therefore, we measured the alignment of population covariability with the sound discrimination axis in both passive and task-engaged states (*Figure 5B*). Surprisingly, in A1, task engagement caused covariability to become more aligned with the sound discrimination axis. These results are consistent with a model in which the principal covariability axis does not represent information limiting noise in early sensory areas (*Kafashan et al., 2021*), but instead reflects top-down, task-dependent gain modulation becoming more aligned with the task-relevant coding axis (*Denfield et al., 2018*; *Goris et al., 2014*; *Rabinowitz et al., 2015*). Conversely, in dPEG alignment was low in both the passive and engaged states, consistent with covariability reflecting non-sensory variables that do not directly interact with the processing of the sensory stimulus (*Stringer et al., 2019a*).

To directly measure how these population-level changes relate to sound representation and emergent behavioral selectivity in non-primary auditory cortex, we performed simulations based on factor analysis model fits in which we sequentially introduced task-dependent changes in mean sound-evoked response gain, single neuron variance, and population covariance matching changes in the actual neural data (Methods: Factor analysis – Simulations). A simulation in which population covariability was fixed and only the evoked response gain changed between passive and active conditions (gain only) was sufficient to produce task-relevant selectivity in non-primary auditory cortex (*Figure 5C–D*). Thus, task-dependent changes in evoked response gain, not population covariability, support the emergence of a behaviorally relevant population code in non-primary auditory cortex.

This result is consistent with the fact that population covariability did not change in a systematic way with respect to the sound discrimination axis in non-primary auditory cortex (*Figure 5B*). However, in A1 this was not the case. Therefore, we hypothesized that in A1 modeling changes in covariability would be required to explain the observed task-dependent changes in generalized sound discriminability. Indeed, we observed monotonic improvement in the model's ability to predict overall Δd-prime in A1, confirming that the shared variability model captured real aspects of shared population covariability that contribute to the accuracy of sound representation in A1 (*Figure 5—figure supplement 2*).

Finally, we used the same simulation approach to determine what aspects of population activity carry the 'choice' related information we observed in A1 and dPEG (*Figure 4—figure supplement 1*). Similar to our findings for stimulus decoding, we found that gain modulation alone was sufficient to recapitulate the choice information present in the raw data for this task. This helps frame prior work that pooled neurons across sessions to study population coding of choice in similar auditory discrimination tasks (*Christison-Lagay et al., 2017*).

## Discussion

We observed distinct changes in how neural populations represent sound categories between primary (A1) and non-primary (dPEG) auditory cortex during a challenging tone-in-noise task. In A1, task engagement improved neural coding uniformly across all sound categories, both relevant and irrelevant to the current task. In dPEG, on the other hand, the neural population selectively enhanced the representation only of sound categories relevant to the tone-in-noise behavior. Task-dependent changes in neural response gain were sufficient to account for this emergent selectivity. In addition, we observed striking changes in population-level correlated activity that were strongly dependent on brain region. The pattern of task-related effects is consistent with a hierarchical, mixed selectivity

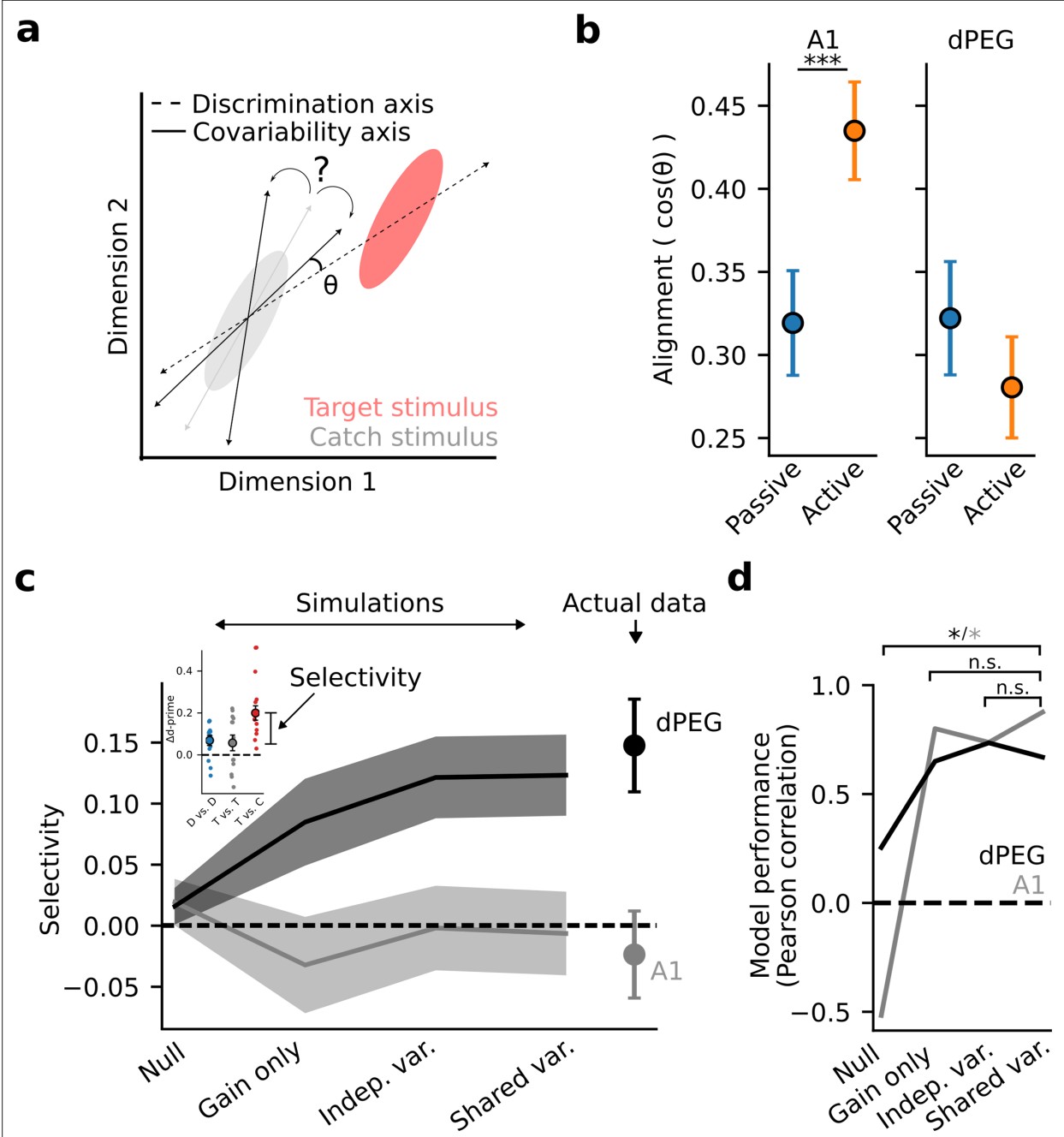

**Figure 5.** Task-related changes in shared population covariability do not impact the coding of task-relevant features. (**a**) Schematic of population response over many trials to a catch stimulus (gray) and target stimulus (red), projected into a low-dimensional space. Dashed line indicates the sensory discrimination axis and the gray line indicates the axis of shared variability across trials during passive listening. Black lines indicate possible rotations in the axis of shared variability either toward or away from the discrimination axis during the task-engaged state. A larger angle ($\theta$) between the shared variability and the discrimination axes leads to increased discrimination accuracy. (**b**) Alignment (cosine similarity) between the discrimination and shared variability axes during passive and active conditions. Error bars represent the standard error of the mean. The axes become more aligned during task engagement in A1 (p<0.001, Wilcoxon signed-rank test) and do not change in dPEG. (**c**) Mean selective enhancement of neural target vs. catch discriminability across recording sites for simulated and actual data. Shading represents the standard error of the mean across experiments (A1: n=18 sessions, n=3 animals, dPEG: n=12 sessions, n=4 animals). Selective enhancement is the difference in Δd-prime for target vs. catch and target vs. target (inset). Simulations sequentially introduced task-dependent changes in mean sound-evoked response gain, single neuron variance, and population covariance matching changes in the actual neural data. (**d**) Model performance is defined as the correlation coefficient between simulated and actual selective enhancement across all sessions. Performance of each model was evaluated against the performance of the shared variance model to check

*Figure 5 continued on next page*

*Figure 5 continued*

for stepwise improvements in predictions. Stars indicate significance at alpha = 0.05 level, bootstrap test. Colors indicate brain regions: dPEG/black, A1 /gray.

The online version of this article includes the following figure supplement(s) for figure 5:

**Figure supplement 1.** Modulation of population covariability metrics by task engagement.

**Figure supplement 2.** Modeling task-dependent changes in shared population covariability improves predictions of decoding changes in A1, but not dPEG.

model of sensory decision-making (*Rigotti et al., 2013*). Neural populations in early brain areas form a sparse, overcomplete representation of sensory inputs, which supports a simple linear readout of task-relevant features in downstream areas. The selective changes are measurable only at the population level in dPEG, but a subsequent stage of processing would support category-specific coding by single neurons, as in frontal cortex (*Fritz et al., 2010*; *Tsunada et al., 2011*).

### Emergent invariant, behaviorally relevant sound representations in non-primary auditory cortex

It has been proposed that auditory cortex dynamically computes representations of task-relevant sound features from non-specific spectro-temporal inputs to A1 (*Lestang et al., 2023*). Prior work has established that task-dependent modulation of auditory responses is larger in non-primary versus primary fields of auditory cortex (*Atiani et al., 2014*; *Kline et al., 2023*; *Niwa et al., 2013*). In our study, we asked how specific these changes in neural activity are to the encoding of task-relevant vs. irrelevant sound features. We found a clear dissociation between cortical fields; behaviorally relevant representations first emerged in the non-primary field, dPEG. In the frontal cortex (FC) of ferrets engaged in a similar tone detection behavior, single-neuron activity is behaviorally gated. Responses to a set of target tones are strongly enhanced when they require a behavioral response (*Fritz et al., 2010*). In our task, the enhanced category representation in dPEG supports a simple linear readout of 'go' versus 'no-go' categories that could provide input to category-specific neurons in FC.

Consistent with the hypothesis that the selective enhancement of category representation is causally related to behavior, we found that decoding accuracy in dPEG tracked the animal's behavioral performance. This correlation between neural activity and behavioral performance is consistent with previous observations that choice-related activity is present or is stronger in non-primary versus primary auditory cortex (*Bizley et al., 2013*; *Tsunada et al., 2016*). However, some other studies have reported choice-related activity emerging as early as A1 (*Niwa et al., 2012b*; *Selezneva et al., 2006*), suggesting that the role of different cortical regions in decision-making may depend on aspects of the task, including the specifics of the auditory stimuli and the associated motor response. Future studies that precisely stimulate or suppress activity along the auditory pathway may definitively probe the causal role that each region plays in auditory perception.

### Population activity reveals hierarchically organized representations in the auditory system

Despite the diverse task-dependent changes in sound representations across individual neurons, analysis of sound discriminability at the population level revealed striking qualitative differences between A1 and dPEG. This finding highlights the value of studying auditory coding at the level of neural populations, which provides a more complete assessment of system-wide function than individual neurons (*Bagur et al., 2018*; *Lestang et al., 2023*). In our analysis, we defined population coding accuracy as the amount of stimulus information an ideal observer could extract from simultaneously recorded population activity. Thus, our results should be interpreted as an upper bound on the information transmitted by a group of neurons about a particular stimulus. Critically, these measures do not necessarily reflect the information utilized by the animal. In A1, for example, we found a decoding axis for every stimulus category along which task engagement improved sound representation. Despite this global improvement in A1, decoding downstream in dPEG was only improved for task-relevant sounds. This selective change indicates that dPEG does not always read out information optimally from A1. Instead, during the engaged state, it reads out activity along an axis of A1 population activity that is invariant to task-irrelevant stimuli.

At face value, these findings may seem paradoxical. If only one dimension of A1 activity is utilized downstream, why does task engagement improve sound representations so broadly? Theories of neural disentanglement and the formation of categorical representations posit that the brain must first build overcomplete, high-dimensional representations of the sensory world (*DiCarlo and Cox, 2007*). From this high-dimensional activity, it is straightforward to build a linear decoder, tuned to the task at hand, that extracts only task-relevant information (*Rigotti et al., 2013*). Our findings are consistent with this theory and describe a hierarchical network for computing sound categories. An overcomplete representation in A1 is selectively filtered at the population level in dPEG, and subsequently, this activity may provide input to category-specific neurons in areas such as FC.

## Implications for the role of correlated activity in sensory processing

An important advantage of our experimental setup was that we simultaneously recorded the activity of populations of neurons, contrasting with previous studies that built pseudo-populations from serial experiments (*Bagur et al., 2018*). This approach allowed us to investigate how trial-by-trial covariability across the population depends on task engagement and contributes to sound encoding. Theoretical work has shown that trial-by-trial covariance can impair population coding accuracy (*Averbeck et al., 2006*). Early experiments in visual cortex supported this idea, demonstrating that selective attention improves perceptual discriminations primarily by reducing covariance (*Cohen and Maunsell, 2009*). We found task engagement modulated covariability patterns in both A1 and dPEG, broadly consistent with prior work in the auditory system (*Downer et al., 2017*).

Strikingly, however, the changes in covariance had no impact on emergent behavioral selectivity in dPEG or on the mean generalized improvement in sound coding in A1. What, then, do these changes in correlated neural activity reflect? In A1, we found that covariability became more aligned with the behaviorally relevant sensory decoding axis during task engagement. This finding is in opposition to a model in which covariability reflects information-limiting noise (*Bartolo et al., 2020*; *Moreno-Bote et al., 2014*; *Rumyantsev et al., 2020*). Instead, we hypothesize that covariability in A1 primarily reflects top-down gain modulation that drives changes in selectivity (*Denfield et al., 2018*; *Goris et al., 2014*; *Guo et al., 2017*). During the task-engaged state, top-down signals selectively modulate sound-evoked responses of neurons tuned to task-relevant stimuli, thus boosting the representation of task-relevant sounds for downstream readout. If gain is not perfectly static, but varies in strength from trial to trial, this could explain the observed increase in covariability amongst task-relevant neurons (*Denfield et al., 2018*). Simultaneous recordings from multiple auditory fields that permit analysis of the communication subspace between areas may provide further insight into the interaction between top-down signals and sound-evoked responses (*Semedo et al., 2019*; *Srinath et al., 2021*).

In contrast, the direction of covariability in dPEG changed randomly with respect to the behaviorally relevant decoding axis. During both passive and engaged states, covariability remained mostly orthogonal to the sensory decoding axis and, therefore, had little impact on population decoding accuracy. These findings are consistent with recent work suggesting that trial-by-trial covariability is primarily orthogonal to sensory coding dimensions and reflects non-sensory motor or cognitive variables, such as whisking, running, or arousal (*Musall et al., 2019*; *Stringer et al., 2019b*). Our results contribute to a growing body of evidence that covariability does not usually reflect information limiting noise, but instead reflects important cognitive processes active in different brain regions during sensory decision-making (*Srinath et al., 2021*).

## Methods
### Surgical procedures
All procedures were approved by the Oregon Health and Science University Institutional Animal Care and Use Committee (IACUC protocol IP1561) and conform to the standard of the Association for Assessment and Accreditation of Laboratory Animal Care (AAALAC). Adult male domestic ferrets were acquired from an animal supplier (Marshall BioResources). To permit head fixation during neurophysiological recordings and behavioral training, all animals underwent head-post implantation surgeries. Surgeries were performed as described previously (*Saderi et al., 2021*; *Slee and David, 2015*). Two stainless steel head-posts were fixed to the skull along the midline with bone cement (Palacos or Charisma). Additionally, 8–10 stainless steel screws were inserted into the skull

and bonded to the bone cement to form the structure of the implant. After a two-week recovery period, animals were slowly habituated to a head-fixed posture and auditory stimulation. Following behavioral training on the tone-in-noise task (see below), a micro craniotomy (0.5–1 mm) was opened above either primary auditory cortex (A1) or the dorsal posterior ectosylvian gyrus (dPEG) to allow for the insertion of neurophysiology recording electrodes. Recording sites were targeted based on external landmarks and tonotopic maps (*Atiani et al., 2014*; *Bizley et al., 2005*). After recordings were complete at one location, the microcraniotomy was allowed to close and a new one was opened at a different location.

## Behavioral paradigm

Four adult male ferrets were trained on a positively reinforced, go/no-go tone-in-noise detection task. Throughout behavioral training, animals were provided with free access to water on weekends and placed on partial water restriction during the week. During restriction periods, animals were only able to receive liquid rewards during behavioral training. Supplemental water was provided after a training session, if necessary, to ensure that animals maintained at least 90% of their baseline body weight throughout training.

Single behavioral trials consisted of a sequence of narrow-band noise bursts (0.3-octave bandwidth, 0.3 s duration, 0.2 s ISI), followed by a target tone (0.3 s duration). Animals reported the presence of the target tone by licking a water spout. Licks were detected through a piezo electric sensor glued to the spout (two animals) or by a video camera monitoring the luminance change in a window around the spout (two animals). Licks occurring during a target window (0.2–1.0 s following target onset) were rewarded with a high-protein, high-calorie supplement (Ensure), while licks outside the window were penalized with a timeout (3–10 s). The number of distractor stimuli per trial was distributed randomly with a flat hazard function to prevent behavioral timing strategies. Each behavioral session consisted of 100–200 trials. A subset of trials contained an explicit catch stimulus – a noise burst with the same center frequency as the target tone and occurring with the same temporal distribution as targets. Trials containing a catch stimulus were always concluded by a pure tone reminder target, which was rewarded if the animal successfully withheld responding to the catch licked in response to the pure tone.

The center frequencies of distractor noise bursts spanned three octaves around the target tone frequency, which was varied randomly between days (0 .l–20 kHz). Initially, training sessions contained only a single pure tone target (Inf dB SNR). As training progressed, masking noise was introduced to the target tone in order to increase task difficulty. By the end of training, a single behavioral session could consist of up to four different target stimuli (–10 dB, –5 dB, 0 dB, Inf dB). More difficult target stimuli (e.g. –10 dB) occurred more rarely than easier stimuli (e.g. Inf dB) during behavioral sessions to maintain motivation. In all cases, noise masking the target was exactly matched to the catch stimulus (centered at the target frequency). Target frequency was fixed within a session, and variable SNR was achieved by adjusting tone amplitude relative to the fixed masking noise. That is, the masking noise (and distractor stimuli) were always presented with an overall sound level of 60 dB SPL. Infinite (inf) dB trials corresponded to trials where the target tone was presented at 60 dB SPL without any masking noise present, 0 dB to trials where the target was 60 dB SPL, –5 dB to trials where the target was presented at 55 dB SPL etc. Neurophysiological recordings proceeded only after animals were able to perform the full, variable SNR task with consistently above chance level performance on –5 dB SNR target tones.

## Acoustic stimuli

All experiments were performed in a sound-attenuating chamber (Gretch-Ken). Sound presentation and behavioral control were provided by custom MATLAB software (https://bitbucket.org/lbhb/baphy). Digital acoustic signals were transformed to analog (National Instruments), amplified (Crown), and delivered through free-field speakers (Manger, 50–35,000 Hz flat gain). Speakers were located 80 cm from the animal at +/-30 deg. azimuth. Stimuli were presented from a single speaker (left or right). During neurophysiology experiments, the speaker contralateral to the recording hemisphere was used. Sound level was equalized and calibrated against a standard reference (PCB Piezoelectronics).

## Neurophysiology

Neurophysiological recordings were performed using 64-channel silicon microelectrode arrays (*Shobe et al., 2015*). Electrode contacts were spaced 20 μm horizontally and 25 μm vertically in three columns, collectively spanning 1.05 mm of cortex. Data were amplified (RHD 128-channel head stage, Intan Technologies), digitized at 30 kHz (Open Ephys *Black et al., 2017*), and saved to disk for offline analysis.

Spike sorting was performed using Kilosort2 (*Pachitariu et al., 2016*), followed by curation in phy (https://github.com/cortex-lab/phy; *Rossant, 2024*). For all identified spike clusters, we quantified isolation as one minus a contamination percentage, defined based on the cluster's isolation in feature space. We categorized spikes with isolation >85% as isolated or nearly isolated units and included them in all analyses in this study.

## Auditory field localization

Initial recordings targeted A1 using external landmarks (*Radtke-Schuller, 2018*). Tuning curves were calculated using pure tone stimuli (100ms duration, 200ms ISI, 3–7oc taves). Neurons were confirmed to be in A1 based on stereotypical response properties: short latency responses to sound onset, sharp and consistent frequency tuning across layers, and a characteristic dorsal-ventral tonotopic map across penetrations (*Shamma et al., 1993*). Once A1 was located, subsequent craniotomies were opened in small lateral steps. Tuning was measured at each recording site, and the border between A1 and dPEG was defined as the location where the tonotopic map gradient reversed (*Atiani et al., 2014*; *Bizley et al., 2005*). After all recording sessions were completed, the best frequencies for each penetration were plotted according to their stereotactic coordinates for post-hoc confirmation of the boundary between A1 and dPEG. Ambiguous recording sites that could not be confidently placed into either area based on their frequency tuning were excluded from the analysis.

## Pupil recording

To account for spontaneous fluctuations in arousal that can modulate cortical activity (*McGinley et al., 2015*; *Schwartz et al., 2020*), infrared video of the animal's eye was recorded for offline analysis (camera: Adafruit TTL Serial Camera, lens: M12 Lenses PT-2514BMP 25.0 mm). The eye ipsilateral to the neurophysiological recording site was recorded so that camera hardware did not interfere with contralateral sound stimuli. To measure pupil size, we fit an ellipse to the boundary of the animal's pupil on each frame using a custom machine-learning algorithm based on DenseNet201 (*Huang et al., 2018*) and save the dimensions of the ellipse on each frame. Pupil size was shifted 750ms relative to spike times in order to account for the previously reported lagged relationship between neural activity and pupil in cortex (*McGinley et al., 2015*).

## Analysis of behavioral performance

Behavioral performance was measured using d-prime (*Green and Swets, 1966*), defined as the z-scored difference between the target hit rate and false alarm rate across a behavior session. We measured the false alarm rate from response to the catch stimulus, whose temporal distribution within a trial was explicitly balanced with target locations across trials. Thus, for each target SNR, d-prime described how well the animal could discriminate that target from the catch stimulus. A d-prime of 0 indicates chance level performance.

## Single neuron evoked activity and d-prime

Responses of single neurons to task stimuli were measured by computing each neuron's peri-stimulus time histogram (PSTH) response to each stimulus. For visualization (e.g. *Figure 2*), spiking activity was binned at 20ms, normalized to its 100ms pre-stimulus baseline, z-scored, and smoothed with a gaussian kernel of width of 30ms. Single-trial responses were computed as a neuron's mean z-scored activity during the 300ms sound evoked window. For active trials, we included responses measured on hit, correct reject, and miss trials. To quantify neural discriminability between catch and target sounds, we measured the difference between the mean z-scored response to target versus catch stimuli (neural d-prime), which is analogous to the behavioral d-prime described above.

## Neural population d-prime

To determine how well the activity of simultaneously recorded populations of neurons could discriminate between task stimuli, we measured neural population d-prime. Similar to the single neuron

metric, population d-prime was defined as the z-scored difference in the population response to two distinct sound stimuli. We projected high-dimensional z-scored population activity onto a one-dimensional optimal linear discrimination axis to compute d-prime (**Abbott and Dayan, 1999**), where $\Delta\mu$ is equal to the mean difference in response to the two stimuli and $\Sigma$ is the stimulus-independent covariance matrix.

$$d'^2 = \Delta\mu^T \Sigma^{-1} \Delta\mu$$

Prior work has shown that finding an optimal discrimination axis for large neural populations can be unreliable because of overfitting to trial-limited data (**Kanitscheider et al., 2015**). To avoid over-fitting, we performed *dDR* prior to computing the discrimination axis (**Heller and David, 2022**). In brief, this procedure projected the population activity into the two-dimensional space spanned by the population covariability axis (noise axis) and sound discrimination axis (signal axis), where $e_1$ is the first eigenvector of the population covariance matrix.

$$signal = \mu_a - \mu_b = \Delta\mu^T$$
$$noise = e_1 - e_1 \Delta\mu^T$$

The full dimensionality reduction and decoding procedure was repeated for each stimulus pair individually to avoid bias from stimuli that produced different magnitude responses. Results were grouped into behaviorally relevant (target versus catch) and behaviorally irrelevant (target versus target, distractor versus distractor) categories. To be included in the analysis, we required that a sound stimulus must have been presented in at least five active and five passive trials. Thus, the number of target stimuli analyzed per session depended on the animal's performance and how long they remained engaged in the task; for shorter behavioral sessions, fewer repetitions of each stimulus were presented and target conditions with low repetition count were omitted.

## Choice probability analysis

We performed a choice decoding analysis on hit vs. miss trials. We followed the same procedure as described above for stimulus decoding, where instead of a pair of stimuli our two classes to be decoded were 'hit trial' vs. 'miss trial.' That is, for each target stimulus we computed the optimal linear discrimination axis separating hit vs. miss trials (**Abbott and Dayan, 1999**) in the reduced dimensionality space identified with dDR (**Heller and David, 2022**). For the sake of interpretability with respect to previous work, we reported choice probability as the percentage of correctly decoded trial outcomes rather than d-prime. Percent correct was calculated by projecting the population activity onto the optimal discrimination axis and using leave-one-out cross-validation to measure the number of correct classifications.

## Factor analysis – population metrics

To characterize population-wide covariability we used factor analysis (**Umakantha et al., 2021**). Factor analysis is the simplest form of dimensionality reduction that explicitly separates shared variance across neurons from independent variance of single neurons, decomposing the spike count covariance matrix into two parts, a covariance matrix representing shared variance between neurons ($\Sigma_{shared}$) and a diagonal matrix representing the independent variance of each single neuron ($\Psi$):

$$\Sigma = \Sigma_{shared} + \Psi$$

Because we were interested in stimulus-independent trial-trial variance and the role it played in behaviorally relevant sound decoding, we performed this analysis only on responses to the catch stimulus, as this was common to all measurements of pairwise target vs. catch discrimination accuracy. This way, spike-count covariance was not due to changing stimulus conditions and we could directly ask how it interacted with the behaviorally relevant, target vs. catch discrimination axis. We fit a unique factor analysis model for each behavior state (active verses passive) and experiment. The number of total factors was selected as the model which maximized log-likelihood. Following prior work, we quantified the properties of each factor analysis model using three metrics (**Umakantha et al., 2021**):

*Loading similarity:* Similarity of neuronal loading weights for the Factor that explained maximal shared variance. A value of 0 indicates maximal dissimilarity of weights and a value of 1 indicates that the weights for all neurons are identical.

*Percent shared variance (%sv):* The percentage of each neuron's variance that can be explained using other neurons in the population. Ranges from 0 to 100%.

*Dimensionality:* The number of dimensions that maximized log-likelihood. In other words, the rank of the shared spike-count covariance matrix. Integer value.

## Factor analysis – simulations

We simulated neural population responses to each target and catch stimulus by drawing samples from a multivariate gaussian distribution (n=2000 responses were generated for each sound/behavior state). The mean response of each neuron was determined using the neuron's actual PSTH and covariance between neurons was defined as the rank $R<N$ covariance matrix that maximized the likelihood of the Factor Analysis model for each sound stimulus. Data were simulated independently for each behavior state (passive listening vs. active task engagement). We simulated activity with four models:

### Null

Mean response, independent variance, and covariance of the gaussian distribution were fixed to the active neuron PSTH, active independent variance ($\Psi_{active}$), and active covariance matrix ($\Sigma_{shared,\,active}$) for both passive and engaged states. Thus, simulated data ($r_{sim}$) were statistically identical between passive and engaged conditions.

$$r_{sim,\,active} = N(\mu_{active}, \Sigma_{shared,\,active} + \Psi_{active})$$

$$r_{sim,\,passive} = N(\mu_{active}, \Sigma_{shared,\,active} + \Psi_{active})$$

### Gain only

Independent variance and covariance for both passive and engaged states were fixed to the active estimates, but the mean was matched to the actual condition's PSTH. Thus, variance was independent of task but mean evoked response magnitude was allowed to be modulated by task engagement.

$$r_{sim,\,active} = N(\mu_{active}, \Sigma_{shared,\,active} + \Psi_{active})$$

$$r_{sim,\,passive} = N(\mu_{passive}, \Sigma_{shared,\,active} + \Psi_{active})$$

### Independent variance

Mean evoked response and independent variance were modulated by task engagement. Covariance was fixed between states.

$$r_{sim,\,active} = N(\mu_{active}, \Sigma_{shared,\,active} + \Psi_{active})$$

$$r_{sim,\,passive} = N(\mu_{passive}, \Sigma_{shared,\,active} + \Psi_{passive})$$

### Shared variance

All parameters of the gaussian distribution were matched to the state-dependent estimates.

$$r_{sim,\,active} = N(\mu_{active}, \Sigma_{shared,\,active} + \Psi_{active})$$

$$r_{sim,\,passive} = N(\mu_{passive}, \Sigma_{shared,\,passive} + \Psi_{passive})$$

## Alignment of population covariability axes

To measure the alignment of two population covariability axes, we used the absolute value of their cosine similarity. Thus, alignment ranged from 0 (perfectly orthogonal) to 1 (perfectly aligned). In the noise axis versus discrimination axis alignment analysis, we defined the noise axis as the Factor that explained the most shared variance in the catch response (See above section: Factor analysis).

## Pupil-indexed arousal control

To control for changes in neural activity that were due to non-specific increases in arousal rather than task engagement (*Saderi et al., 2021*), we used linear regression to remove variability in the activity of single neurons that could be explained by pupil size. The response of each neuron to each stimulus, $r_i(t)$, was modeled as a linear function of pupil size, $p(t)$:

$$\widehat{r_i}(t) = \alpha_i\, p(t) + \beta_i$$

Then, to remove the pupil-explainable variance from each neuron's response but preserve any pupil-independent effect of task engagement on activity, we defined the corrected firing rate, $\bar{r}_i(t)$, as the true response minus the pupil-dependent portion of the regression model:

$$\bar{r}_i(t) = r_i(t) - \alpha_i\, p(t)$$

Thus, the mean sound evoked response was preserved but changes correlated with pupil were removed. This procedure was performed prior to the analysis of task-dependent selectivity in dPEG (e.g. *Figure 3*). Results were similar for a control analysis that ignored pupil-dependent changes, indicating that the emergent selectivity in dPEG does not depend on this correction for generalized effects arousal (*Figure 3—figure supplement 2*).

## Statistical tests

Given that the noise properties of neurons are not well understood, particularly around changes in behavioral state, a formal power analysis was not possible. Choice of sample size was based on standard practices in studies of cortex in ferrets and similarly sized species. For each experimental recording session, we measured the population decoding performance of multiple stimulus pairs. To control for any statistical dependencies between these data points within a recording session, we first took the mean projection across stimulus pairs within each recording session before measuring p-values. This procedure reduces our statistical power but provides more conservative estimates of statistical significance which are more robust to detecting spurious false positive results. For all pair-wise statistical tests shown in *Figures 3 and 5*, and *Figure 3—figure supplement 1* we performed a Wilcoxon signed rank test. Significance was determined at the alpha = 0.05 level. The number of unique recording sessions and animals that went into each comparison are listed in the main text/figure legends, along with the p-value for each analysis.

The one exception to this general approach is in *Figure 2*, where we analyzed the sound discrimination abilities of single neurons. In this case, we computed p-values for each neuron and stimulus independently. First, for each neuron and catch vs. target stimulus pair, we measured d-prime (see Methods: Single neuron evoked activity and d-prime). We generated a null distribution of d-prime values for each neuron-stimulus pair, under each experimental condition by shuffling stimulus identity across trials before computing d-prime (100 resamples). A neuron was determined to have a significant d-prime for a given target vs. catch pair if its actual measured d-prime was greater than the 95th percentile of the null d-prime distribution. Second, for each neuron and catch vs. target stimulus pair, we tested if d-prime was significantly different between active and passive conditions. To test this, we followed a similar procedure as above, however, rather than shuffle stimulus identity, we shuffled active vs. passive trial labels. This allowed us to generate a null distribution of active vs. passive d-prime difference for each neuron and stimulus pair. A neuron was determined to have a significant change in d-prime between conditions if the actual Δ d-prime lay outside the 95% confidence interval of the null Δ d-prime distribution.

In *Figure 4*, we tested if the change in neural population d-prime was correlated with behavior performance on a per-target stimulus basis. Because each target had different behavioral performance (due to varying SNR), here we treated each stimulus as an independent sample. Therefore, correlation was measured between n sessions x n target stimuli neural vs. behavioral d-primes. To determine the significance of correlation in each brain region, we performed random bootstrap resampling to generate a null distribution of correlation values. The correlation for a given brain region was deemed significant if the actual observed correlation was greater than the 97.5-percentile of the null distribution.

To evaluate the performance of FA model simulations in predicting behavioral selectivity (*Figure 5*) and Δ d-prime (*Figure 5—figure supplement 2*), we measured the correlation between simulated and actual metrics for each model. To determine if stepwise changes in the FA model (e.g. adding task-dependent gain modulation) caused significant improvements in model performance, we compared correlation coefficients for each model to the correlation coefficient for the final model. To do this, we computed 1000 bootstrap resamples of the correlation coefficient for each model. If the 97.5-percentile of this distribution was greater than the observed correlation for the full model, we determined that it was not significantly different. That is, if the observed correlation for the full model lay within the 95% confidence interval of the null bootstrapped distribution for a given model, it was determined to not be significantly different than the full model.

# Additional information

### Funding

| Funder | Grant reference number | Author |
| --- | --- | --- |
| National Science Foundation | GVPRS0015A2 | Charles R Heller |
| National Institutes of Health | DC0495 | Stephen V David |

The funders had no role in study design, data collection and interpretation, or the decision to submit the work for publication.

### Author contributions

Charles R Heller, Conceptualization, Data curation, Software, Formal analysis, Funding acquisition, Visualization, Writing - original draft, Writing - review and editing; Gregory R Hamersky, Data curation, Writing - review and editing; Stephen V David, Conceptualization, Software, Funding acquisition, Writing - original draft, Project administration, Writing - review and editing

### Author ORCIDs
Charles R Heller ⓘ https://orcid.org/0000-0002-9048-1201
Stephen V David ⓘ https://orcid.org/0000-0003-4135-3104

### Ethics

This study was performed in accordance with the Oregon Health and Science University Institutional Animal Care and Use Committee (IACUC) and conforms to standards of the Association for Assessment and Accreditation of Laboratory Animal Care (AAALAC). (IACUC protocol IP1561).

Reviewer #1 (Public Review): https://doi.org/10.7554/eLife.89936.3.sa1
Reviewer #2 (Public Review): https://doi.org/10.7554/eLife.89936.3.sa2
Author response https://doi.org/10.7554/eLife.89936.3.sa3

# Additional files

### Supplementary files
• MDAR checklist

### Data availability
All data for generating figures are available in the associated data repository. All source code for reproducing the analyses and figures in the manuscript can be found on GitHub, along with instructions for downloading the data from Dryad: https://github.com/crheller/eLife2024_Task, copy archived at *Heller, 2024*.

The following dataset was generated:

| Author(s) | Year | Dataset title | Dataset URL | Database and Identifier |
|-----------|------|---------------|-------------|-------------------------|
| Heller CR, Hamersky GR, David SV | 2024 | Data from: Task-specific invariant representation in auditory cortex | https://datadryad.org/stash/dataset/doi:10.5061/dryad.z08kprrp4 | Dryad Digital Repository, 10.5061/dryad.z08kprrp4 |

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
