## [Editor Report · eLife assessment]

This **important** study provides insights into how the brain constructs categorical neural representations during a difficult auditory target detection task. Through recordings of simultaneous single-unit activity in primary and secondary auditory areas, **compelling** evidence is provided that categorical neural representations emerge in a secondary auditory area, i.e., PEG. The study is of interest to neuroscientists and can also potentially shed light on human psychological studies.

---

## [Referee Report · Reviewer #1 (Public Review)]

This is a very interesting paper which addresses how auditory cortex represents sound while an animal is performing an auditory task. The study involves psychometric and neurophysiological measurements from ferrets engaged in a challenging tone in noise discrimination task, and relates these measurements using neurometric analysis. A novel neural decoding technique (decoding-based dimensionality reduction or dDR, introduced in a previous paper by two of the authors) is used to reduce bias so that stimulus parameters can be read out from neuronal responses.

The central finding of the study is that, when an animal is engaged in a task, non-primary auditory cortex represents task-relevant sound features in a categorical way. In primary cortex, task engagement also affects representations, but in a different way - the decoding is improved (suggesting that representations have been enhanced), but is not categorical in nature. The authors argue that these results are compatible with a model where early sensory representations form an overcomplete representation of the world, and downstream neurons flexibly read out behaviourally relevant information from these representations.

I find the concept and execution of the study very interesting and elegant. The paper is also commendably clear and readable. The differences between primary and higher cortex are compelling and I am largely convinced by the authors' claim that they have found evidence that broadly supports a mixed selectivity model of neural disentanglement along the lines of Rigotti et al (2013). I think that the increasing body of evidence for these kinds of representations is a significant development in our understanding of higher sensory representations. I also think that the dDR method is likely to be useful to researchers in a variety of fields who are looking to perform similar types of neural decoding analysis.

---

## [Referee Report · Reviewer #2 (Public Review)]

This study compares the activity of neural populations in the primary and non-primary auditory cortex of ferrets while the animals actively behaved or passively listened to a sound discrimination task. Using a variety of methods, the authors convincingly show differential effects of task engagement on population neural activity in primary vs non-primary auditory cortex; notably that in the primary auditory cortex, task-engagement (1) improves discriminability for both task-relevant and non-task relevant dimensions, and (2) improves the alignment between covariability and sound discrimination axes; whereas in the non-primary auditory cortex, task-engagement (1) improves discriminability for only task-relevant dimensions, and (2) does not affect the alignment between covariability and sound discrimination axes. They additionally show that task-engagement changes in gain can account for the selectivity noted in the discriminability of non-primary auditory neurons. They also admirably attempt to isolate task-engagement from arousal fluctuations, by using fluctuations in pupil size as a proxy for physiological arousal. This is a well-carried out study with thoughtful analyses which in large part achieves its aims to evaluate how task-engagement changes neural activity across multiple auditory regions . As with all work, there are several caveats or areas for future study/analysis. First, the sounds used here (tones, and narrow-band noise) are relatively simple sounds; previous work suggests that exactly what activity is observed within each region (e.g., sensory only, decision-related, etc) may depend in part upon what stimuli are used. Therefore, while the current study adds importance to the literature, future work may consider the use of more varied stimuli. Second, the animals here were engaged in a behavioral task; but apart from an initial calculation of behavioral d', the task performance (and its effect on neural activity) is largely unaddressed.

---

## [Author Response]

The following is the authors’ response to the original reviews.

**Public Reviews:**

**Reviewer #1 (Public Review):**
…I find the concept and execution of the study very interesting and elegant. The paper is also commendably clear and readable. The differences between primary and higher cortex are compelling and I am largely convinced by the authors' claim that they have found evidence that broadly supports a mixed selectivity model of neural disentanglement along the lines of Rigotti et al (2013). I think that the increasing body of evidence for these kinds of representations is a significant development in our understanding of higher sensory representations. I also think that the dDR method is likely to be useful to researchers in a variety of fields who are looking to perform similar types of neural decoding analysis.

Thanks! We agree that questions around population coding and high-level representations are critical in the field of sensory systems.

**Reviewer #2 (Public Review):**
... This is a well-carried out study with thoughtful analyses which in large part achieves its aims to evaluate how task-engagement changes neural activity across multiple auditory regions. As with all work, there are several caveats or areas for future study/analysis. First, the sounds used here (tones, and narrow-band noise) are relatively simple sounds; previous work suggests that exactly what activity is observed within each region (e.g., sensory only, decision-related, etc) may depend in part upon what stimuli are used. Therefore, while the current study adds importantly to the literature, future work may consider the use of more varied stimuli. Second, the animals here were engaged in a behavioral task; but apart from an initial calculation of behavioral d', the task performance (and its effect on neural activity) is largely unaddressed.

The reviewer makes several important points that we hope we addressed in the specific changes detailed below. Indeed, it is important to recognize the possibility that the specific stimuli involved in a task may interact with the effects of behavioral state and that variability in task performance should be considered as an important aspect of behavioral state.

**Reviewer #1 (Recommendations For The Authors):**
I have a few minor comments and criticisms:(1) Figure 1c. The choice of low-contrast grey text (e.g. "Target vs. target" is unfortunate, especially when printed), and should be replaced (e.g. with dark grey).

We have edited the figure to use a higher contrast (dark grey). Thanks for catching this.

(2) Figure 2 and Supplementary Figure 3. I think some indication of error or significance is required in all panels. Without this, it's hard to interpret any of these panels.

Thank you for this feedback. Including significance here was clarifying and helps to strengthen our claim that state-dependent changes in neural activity were smaller and more diverse for single neurons than at the population level. We modified Figure 2b-c to indicate whether each neuron’s response to the target stimulus was significantly different than its response to the catch stimulus. The same test was performed in Supplementary Figure 3. Additionally, we added a statistical test in Figure 2d-e to indicate, for each pair of target/catch stimuli, whether discrimination (d-prime) changed significantly between active and passive conditions. Furthermore, we modified the text of the second paragraph under the results heading: “Diverse effects of task engagement on single neurons in primary and non-primary auditory cortex” to reference and interpret the results of these significance tests. The new text reads as follows (L. 121):

“Sound-evoked spiking activity was compared between active and passive states to study the impact of task engagement on sound representation. In both A1 and dPEG, responses to target and catch stimuli were significantly discriminable for a subset of single neurons (about 25% in both areas, Figure 2A-C, Supplemental Figures 3-5, bootstrap test). This supports the idea that stimulus identity can be decoded in both brain regions, regardless of task performance. However, the fact that the responses of most neurons in both brain areas could not significantly discriminate target vs. catch stimuli also highlights the diversity of sound encoding observed at the level of single neurons. The accuracy of catch vs. target discrimination for each neuron was quantified using neural d-prime, the z-scored difference in target minus catch spiking response for each neuron (Methods: Single neuron PSTHs and d-prime (Niwa et al., 2012a)). Task engagement was associated with significant changes in catch vs. target d-prime for roughly 10% of neurons in both A1 (40 / 481 neurons, bootstrap test) and dPEG (33 / 377 neurons, bootstrap test). This included neurons that both increased their discriminability and decreased their discriminability (Figure 2D-E). Thus, the effects of task engagement at the level of single neurons were relatively mild and inconsistent across the population; many neurons showed no significant change and of those that did, effects were bidirectional (Figure 2D-E).”

We also included an additional methods paragraph in the “Statistical tests” section to describe the bootstrapping procedure used for these significance tests (L. 644):

“The one exception to this general approach is in Figure 2, where we analyzed the sound discrimination abilities of single neurons. In this case, we computed p-values for each neuron and stimulus independently. First, for each neuron and catch vs. target stimulus pair, we measured d-prime (see Methods: Single neuron evoked activity and d-prime). We generated a null distribution of d-prime values for each neuron-stimulus pair, under each experimental condition by shuffling stimulus identity across trials before computing d-prime (100 resamples). A neuron was determined to have a significant d-prime for a given target vs. catch pair if its actual measured d-prime was greater than the 95th percentile of the null d-prime distribution. Second, for each neuron and catch vs. target stimulus pair, we tested if d-prime was significantly different between active and passive conditions. To test this, we followed a similar procedure as above, however, rather than shuffle stimulus identity, we shuffled active vs. passive trial labels. This allowed us to generate a null distribution of active vs. passive d-prime difference for each neuron and stimulus pair. A neuron was determined to have a significant change in d-prime between conditions if the actual Δ d-prime lay outside the 95% confidence interval of the null Δ d-prime distribution.”

For Figure 2a, we chose not to indicate significance on the figure to avoid clutter, since the significance for all neurons in the population are shown in panels b-c anyway. Additionally, the difference plot shown in panel a is in units of z-scores, which we believe already gives a raw sense of the significance of the target vs. catch response change per neuron in this example dataset.

(3) Figure 2 and Supplementary Figure 3. I would consider including some more examples as a Supplementary Figure (and perhaps combining Supp Fig 3 with Fig 2 as a main figure).

We found no significant or apparent difference in single-neuron properties between A1 and dPEG. Therefore, we decided it is not helpful to plot both A1 and PEG examples in the main text. However, we agree that the ability to see more examples of the raw data could be useful. Therefore, we compiled two supplementary figures (Supplementary Figures 4 and 5) that replicate Figure 2a for all datasets, encompassing A1 and PEG.

(4) Figure 2a and Supp Fig 3a. I was initially confused that the "delta-spk/sec (z-score)" values had themselves been z-scored, but now I think that they are simply the differences of the two left hand sub-panels. This could be made clear in the figure legend.

The figure legends have been modified to state the procedure for computing “delta-spk/sec” more clearly. Specifically, we added the following information to the legend (L. 141):

“Difference is computed as the z-scored response to the target minus the z-scored catch response (resulting in a difference shown in units of z-score).”

(5) Figure 2b-e and Supp Fig 3b-e. Indicate the time window over which the responses were measured, and the number of neurons.

Figure legends have been modified to include a sentence clearly stating the time window over which responses were measured. The number of neurons is also now included in the legend and on the figure itself. Furthermore, a brief description of the new statistical testing procedure has been added here (L. 144).

“Responses were defined as the total number of spikes recorded during the 300 ms of sound presentation (area between dashed lines in panel A). Neurons with a significantly different response to the catch vs. target stimulus are indicated in black and quantified on the respective figure panel.”

(6) Figure 2. "singe" should read "single"

Typo in figure label has been fixed.

(7) Line 144. Figure number is missing (Figure 3B-C).

The missing figure number has been added to the text.

(8) Figure 3. Again, the low-contrast grey should be replaced.

The low-contrast grey has been replaced with dark grey.

**Reviewer #2 (Recommendations For The Authors):**
This study really nicely compares the activity and effects on activity in two areas of the auditory cortex in respect to task-engagement; I think it is, for the most part, very well done.A couple of specific recommendations:(1) Although I understand 'inf dB' as the SNR, including the actual dB level used in the experiments, would be useful, especially in the case of the inf dB.

Thank you for this feedback. We agree that clarification about the overall sound level used here would be helpful. We have modified the methods section “Behavioral paradigm” to include the following sentence (L. 450):

“That is, the masking noise (and distractor stimuli) were always presented with an overall sound level of 60 dB SPL. Infinite (inf) dB trials corresponded to trials where the target tone was presented at 60 dB SPL without any masking noise present, 0 dB to trials where the target was 60 dB SPL, -5 dB to trials where the target was presented at 55 dB SPL etc.”

In addition, we have modified the main text (L. 82):

“Animals reported the occurrence of a target tone in a sequence of narrowband noise distractors by licking a piezo spout (Figure 1A, Methods: Behavioral paradigm, distractor stimulus sound level: 60 dB SPL). … We describe SNR as the overall SPL of the target relative to distractor noise level. Thus, an SNR of –5 dB corresponds to a target level of 55 dB SPL while an Inf dB SNR corresponds to a target tone presented without any masking noise.”

And Figure legend 1 now explicitly states the sound level used in the experiments (L. 104):

“Variable SNR was achieved by varying overall SPL of the target relative to the fixed (60 dB SPL) distractor noise, e.g., -5 dB SNR corresponds to a 55 dB SPL target with 60 dB SPL masking noise. Infinite (inf) dB SNR corresponds to a target tone presented in isolation (60 dB SPL).”

(2) I very much appreciate the attempt to disentangle task engagement from generalized arousal state, and specifically, addressing this through the use of pupillometry. However, by focusing the discussion of pupil dynamics solely on the arousal-state aspects of pupil size, the paper doesn't address the increasing evidence suggests that pupil size may fluctuate based upon a lot of other things, including perceptual events (see Kronemer et al, 2022 for a recent human paper; for auditory: Zekveld et al 2018 (review) and Montes-Lourido et al, 2021; but many many others, too). It would be nice to see either a bit more nuanced discussion of what pupil size may be indicating (easier), or analyzing the behavior in the context of pupil dynamics (a heavier lift).

This is a good point. We agree that it is worth mentioning these more nuanced aspects of cognition that may be reflected by pupil size. Therefore, we also analyzed pupil size in the context of behavioral performance (see Supplemental Figure 6) and added the following text to the results (L. 193).

“In addition to reflecting overall arousal level, pupil size has also been reported to reflect more nuanced cognitive variables such as, for example, listening effort (Zekveld et al., 2014). Furthermore, rodent data suggests that optimal sensory detection is associated with intermediate pupil size (McGinley et al., 2015), consistent with the hypothesis of an inverted-U relationship between arousal and behavioral performance (Zekveld et al., 2014). To determine if this pattern was true for the animals in our task, we measured the dynamics of pupil size in the context of behavioral performance. Across animals, task stimuli evoked robust pupil dilation that varied with trial outcome (Supplemental Figure 6b-c). Notably, pre-trial pupil size was significantly different between correct (hit and correct reject), hit, and miss trials (Supplemental Figure 6b-c), recapitulating the finding of an inverted-U relationship to performance in rodents (McGinley et al., 2015). Since we focused only on correct trials in our decoding analysis, these outcome-dependent differences in pupil size are unlikely to contribute to the emergent decoding selectivity in dPEG.”

(3) I think it would make this paper shine that much more if behavioral performance were not subsumed into the overall label of task engagement. You've already established you have performance that varies as a function of SNR; I would love to see the neural d' and covariability related to the behavioral d' (in the comparisons where this is possible). I would also love to see a more direct measure of choice for those stimuli that show variable behavior (e.g., a choice probability analysis or something of the like would seem to be easily applied to the target SNRs of -5 and 0 dB); and compare task engaged activity of hits vs misses vs passive listening to those same stimuli. You discuss previous studies looking at choice-related/decision-related activity and draw parallels to this work-given that there is the opportunity with this data set to *directly* assess choice-related activity, the absence of such an analysis seems like a missed opportunity.

Thank you for this feedback. We agree that “task engagement” is not a unimodal state and that a more fine-grained analysis of task-engaged neural activity, according to behavioral choice, could be informative.

First, we would like to point out that in Figure 4 we did already compare behavioral d’ to delta neural d’. We found that the two were significantly correlated in dPEG, but not in A1. This suggests that task-dependent changes in stimulus decoding in dPEG, but not A1, are predictive of behavioral performance. This is consistent with the finding that task-relevant stimulus representations were selectively enhanced in dPEG, but not in A1.

Second, we added a choice decoding analysis to address whether auditory cortex represents the animal’s choice in our task. The results of this analysis are summarized in Supplemental Figure 8 and are discussed under the results section: “Behavioral performance is correlated with neural coding changes in non-primary auditory cortex only.” (L. 226):

“The previous analysis suggests that the task-dependent increase in stimulus information present in dPEG population activity is predictive of overall task performance. Next, we asked whether the population activity in either brain region was directly predictive of behavioral choice on single hit vs. miss trials. To do this, we conducted a choice probability analysis (Methods). We found that in both brain regions choice could be decoded well above chance level (Supplemental Figure 8). Choice information was present throughout the entire trial and did not increase during the target stimulus presentation. This suggests that the difference in population activity primarily reflects a cognitive state associated with the probability of licking on a given trial, or “impulsivity” rather than “choice.” This interpretation is consistent with our finding that baseline pupil size on each trial is predictive of trial outcome (Supplemental Figure 6b).”

To keep our decoding approach consistent throughout the manuscript, we followed the same approach for choice decoding as we did for stimulus decoding (perform dDR then calculate neural d-prime in the dimensionality reduced space). To make the results more interpretable, we converted choice d-prime to a choice probability (percent correctly decoded choices) using leave-one-out cross validation. (We note that d-prime and percent correct are very highly correlated statistics.) This is described in the methods as follows (L. 550):

“We performed a choice decoding analysis on hit vs. miss trials. We followed the same procedure as described above for stimulus decoding, where instead of a pair of stimuli our two classes to be decoded were “hit trial” vs. “miss trial”. That is, for each target stimulus we computed the optimal linear discrimination axis separating hit vs. miss trials (Abbott and Dayan, 1999) in the reduced dimensionality space identified with dDR (Heller and David, 2022). For the sake of interpretability with respect to previous work we reported choice probability as the percentage of correctly decoded trial outcomes rather than d-prime. Percent correct was calculated by projecting the population activity onto the optimal discrimination axis and using leave-one-out cross validation to measure the number of correct classifications.”

(4) It would also be interesting to look at population coding across sessions (although the point is taken that within a session allows the opportunity to assess covariability). Minorly self-servingly but very much related to the above point, Christison-Lagay et al, 2017 employed a similar detect-in-noise task, analyzed single neurons and population level activity, and looked at putative choice-related activity. The current study has the opportunity to expand on that kind of analysis that much more by looking across multiple sites vs within a given recording site; and compare across regions.

Thank you for highlighting this point, we agree that it is important. When studying population coding it is critical to consider the impact of covariability between neurons. Therefore, it is worthwhile to revisit our interpretations of prior results, e.g., Christison-Lagay et al, 2017, which studied population coding by combining neurons across different sessions, given that we now have access to simultaneously recorded population data.

First, we would like to point out that this was the primary motivation for our simulation analyses presented in Figure 5. Using simulations, we found that task-dependent gain modulation (which can be observed across sessions) was sufficient to explain our primary finding – selective enhancement in decoding of behaviorally relevant sound stimuli in dPEG.

Second, to address the question about how covariability affects choice-related information in auditory cortex and compare our findings with prior studies, we performed the same set of simulations for choice probability analysis. We found that, again, choice-dependent gain modulation was sufficient to explain our findings. That is, simulations with hit- vs. miss-dependent gain changes, but fixed covariability, closely mirrored the choice probability we observed in the raw data. An additional simulation where covariability between all neurons was set to zero also recapitulated our findings in the raw data. Collectively, this suggests that covariability does not play a significant role in shaping the choice information present in A1 and dPEG during this task. We have added the following text to the manuscript to summarize this finding (L. 293):

“Finally, we used the same simulation approach to determine what aspects of population activity carry the “choice” related information we observed in A1 and dPEG (Figure 4 – figure supplement 1). Similar to our findings for stimulus decoding, we found that gain modulation alone was sufficient to recapitulate the choice information present in the raw data for this task. This helps frame prior work that pooled neurons across sessions to study population coding of choice in similar auditory discrimination tasks (Christison-Lagay et al, 2017).”